# Tumor-Treating Fields in Glioblastomas: Past, Present, and Future

**DOI:** 10.3390/cancers14153669

**Published:** 2022-07-28

**Authors:** Xiaopeng Guo, Xin Yang, Jiaming Wu, Huiyu Yang, Yilin Li, Junlin Li, Qianshu Liu, Chen Wu, Hao Xing, Penghao Liu, Yu Wang, Chunhua Hu, Wenbin Ma

**Affiliations:** 1Department of Neurosurgery, Peking Union Medical College Hospital, Chinese Academy of Medical Sciences and Peking Union Medical College, Beijing 100730, China; guoxiaopeng_pumch@163.com (X.G.); garywu12345@163.com (J.W.); brenda0316@126.com (H.Y.); yilin_li@student.pumc.edu.cn (Y.L.); pumclijl18@163.com (J.L.); qsliu@student.pumc.edu.cn (Q.L.); wuchen09@126.com (C.W.); xinghao1026@outlook.com (H.X.); liuph14@hotmail.com (P.L.); mawb2001@hotmail.com (W.M.); 2National Engineering Laboratory for Neuromodulation, School of Aerospace Engineering, Tsinghua University, Beijing 100084, China; yangx20@mails.tsinghua.edu.cn; 3Clinical Medicine, Peking Union Medical College, Beijing 100730, China

**Keywords:** glioblastoma, GBM, tumor-treating fields, TTFields, mechanism of action, clinical trial, overall survival, progression-free survival

## Abstract

**Simple Summary:**

Glioblastoma (GBM) is the most common malignant primary brain tumor. Although the standard of care, including maximal resection, concurrent radiotherapy with temozolomide (TMZ), and adjuvant TMZ, has largely improved the prognosis of these patients, the 5-year survival rate is still < 10%. Tumor-treating fields (TTFields), a noninvasive anticancer therapeutic modality, has been rising as a fourth treatment option for GBMs, as confirmed by recent milestone large-scale phase 3 randomized trials and subsequent real-world data, elongating patient overall survival from 16 months to 21 months. However, the mechanisms of antitumor efficacy, its clinical safety, and potential benefits when combined with other treatment modalities are far from completely elucidated. As an increasing number of studies have recently been published on this topic, we conducted this updated, comprehensive review to establish an objective understanding of the mechanism of action, efficacy, safety, clinical concerns, and future perspectives of TTFields.

**Abstract:**

Tumor-treating fields (TTFields), a noninvasive and innovative therapeutic approach, has emerged as the fourth most effective treatment option for the management of glioblastomas (GBMs), the most deadly primary brain cancer. According to on recent milestone randomized trials and subsequent observational data, TTFields therapy leads to substantially prolonged patient survival and acceptable adverse events. Clinical trials are ongoing to further evaluate the safety and efficacy of TTFields in treating GBMs and its biological and radiological correlations. TTFields is administered by delivering low-intensity, intermediate-frequency, alternating electric fields to human GBM function through different mechanisms of action, including by disturbing cell mitosis, delaying DNA repair, enhancing autophagy, inhibiting cell metabolism and angiogenesis, and limiting cancer cell migration. The abilities of TTFields to strengthen intratumoral antitumor immunity, increase the permeability of the cell membrane and the blood–brain barrier, and disrupt DNA-damage-repair processes make it a promising therapy when combined with conventional treatment modalities. However, the overall acceptance of TTFields in real-world clinical practice is still low. Given that increasing studies on this promising topic have been published recently, we conducted this updated review on the past, present, and future of TTFields in GBMs.

## 1. Introduction

Glioblastoma (GBM) is the most malignant type of primary brain tumor, with an extremely dismal 5-year postdiagnosis survival rate of <10% [1,2]. The annual age-adjusted incidence rate of GBM in the United States is 3.23 per 100,000 people [1]. Since the milestone phase 3 European Organization for Research and Treatment of Cancer (EORTC) study published in 2005 by Stupp et al. [3], maximal safe resection followed by concurrent radiotherapy with temozolomide (TMZ, 75 mg/m^2^) and subsequent adjuvant TMZ (150–200 mg/m^2^) has been adopted as the standard-of-care protocol worldwide for newly diagnosed GBM (ndGBM) patients. However, even with these multimodal therapies, GBM remains incurable, with a recurrence rate of 100%. Patient prognosis is bleak, with a median overall survival (OS) of 14.6 months to 16.0 months and progression-free survival (PFS) of only 4.0 months [3,4,5]. Considering the rapid development of treatment modalities and successful improvement of patient prognosis with other solid malignancies, including pancreatic adenocarcinoma and mesothelioma, GBM has, unfortunately, become the most lethal type of human cancer [6].

Tumor-treating fields (TTFields) is a noninvasive treatment modality that applies low-intensity, intermediate-frequency, alternating electric fields over the regions of the body where tumors are localized. The use of TTFields inhibits mitosis and the cell cycle, induces cancer cell autophagy, disturbs DNA repair, undermines cell migration, and thus suppresses tumor growth and invasion [7,8,9,10,11,12]. TTFields therapy also ablates the primary cilia on GBM cells that promote tumor growth and chemoresistance to TMZ and induces nuclear envelope disruption and the subsequent release of naked micronucleus clusters, which activate several types of inflammasomes to induce anticancer immunity in GBMs [13,14,15]. The Food and Drug Administration (FDA) of the United States approved the use of TTFields for the treatment of recurrent GBMs (rGBMs) in 2011 and for ndGBMs in 2015 due to the promising results that it had comparable effects with the use of the physician’s best choice (PBC) for rGBMs (EF-11) and promoted improved survival relative to the standardized Stupp protocol for ndGBMs (EF-14) [4,5,16]. In recent years, the effectiveness and safety of TTFields in treating GBMs have been confirmed in various observational and randomized studies, and it has been established as the fourth treatment option in addition to surgery, radiotherapy, and chemotherapy [17]. TTFields therapy has been granted the “category 1” recommendation for the treatment of ndGBMs by the National Comprehensive Cancer Network (NCCN) guidelines, as well as the “category 2B” recommendation for the treatment of rGBMs.

However, it is undeniable that the use of TTFields in real-world clinical practice remains infrequent (<12% in ndGBMs and <16% in rGBMs) [9]. Debates are ongoing regarding the underlying mechanisms of action and clinical use of TTFields in GBMs. The mechanisms of antitumor efficacy, safety, and clinical benefits when combined with other treatment modalities are far from completely elucidated. Given that increasing studies on this topic have been published recently, we conducted this updated review.

## 2. Mechanisms of Action Underlying the Effects of TTFields

### 2.1. Electromagnetism and the Impact of Electric Intensity and Frequency

#### 2.1.1. Electromagnetism of TTFields

Biological tissues have dielectric properties and conductive properties, and applied electrical stimulation can excite electric fields in biological tissues, thus affecting the physiological activities of tissues. Low-frequency electrical stimulation mainly affects the cell membrane potential of excitable tissues, causing depolarization to produce action potentials, such as during nerve electrical stimulation and cardiac pacing [18,19,20,21,22]. High-frequency electrical stimulation mainly generates dielectric loss, causing significant heating effects in tissues. It is commonly used in radiofrequency tumor ablation and other contexts [23,24], whereas medium-frequency electrical stimulation, such as that performed during the use of tumor-treating electric fields, mainly relies on the electric field force, which affects the movement of polar molecules and macromolecules in cells through an uneven electric field and interferes with mitosis and other processes of cells [25].

Devices for TTFields therapy for gliomas rely on an external array of two pairs of electrodes to excite an electric field in the tissue of the human head, as shown in Figure 1a.

TTFields relies on two main electromagnetic mechanisms to produce effects:

(1) Electric field force: The motion of electrodynamic molecules, such as microtubulin subunits, is affected by the electric field, as shown in Figure 1b. In the electric field, the electrodynamic molecules are pulled due to the force of the electric field and tend to be positioned parallel to the direction of the electric field and thus cannot be aligned properly. Based on Ohm’s law and a simplified version of Maxwell’s equations under quasistatic conditions, the distribution of the electric field in the tissue can be calculated using the following equation:J = σE(1)
E = −∇V(2)
where J is the current density, σ is the electrical conductivity, E is the field strength, and V is the electric potential.

(2) Dielectrophoresis: At the end of the cell division process, the intracellular electric field is not uniformly distributed, and the density of the electric field in the cleavage furrow is higher, which induces dielectrophoresis (DEP), causing polar macromolecules and organelles to move toward the region of the cleavage furrow under the action of electrophoretic forces, as shown in Figure 1c; these forces are calculated as [11]:F_DEP_(ω) = 2πε_m_r^3^Re[(ε_p_ * − ε_m_ *)/(ε_p_ * + 2ε_m_ *)]∇|E_rms_|^2^(3)
where ε_m_ is the absolute dielectric constant of the liquid in which the dielectrophoresis operation is performed, r is the radius of the spherical particles, and E_rms_ is the root mean square value of the applied electric field. The complex dielectric constant in the above equation can be expressed as:ε* = ε − jσ/ω(4)

Where σ is the conductivity, and ε is the dielectric constant. In Equation (3), the subscripts *p* and m denote that in the particle and liquid media, respectively, and ω is the frequency of the applied electric field. The magnitude of the dielectrophoretic force is frequency-dependent, the tumor treatment electric field is also frequency selective, and the two may be correlated.

#### 2.1.2. Impact of Electric Intensity and Frequency on the Efficacy of TTFields

TTFields-induced cancer cell death is dose-dependent; frequency-dependent; and associated with tumor position, tissue homogeneity, and the conductivity distribution of surrounding tissues [25,26,27,28]. High electric density is associated with longer survival and improved quality of life (QoL), whereas low intensity outside the electric fields leads to possible tumor progression [29,30,31]. Longitudinal monitoring of the tumor and prompt relocation of electric arrays are essential to controlling disease progression. Positioning of the electric arrays should be personalized to ensure maximum field intensity at the tumor bed. Skull thinning and the formation of skull burr holes over the tumor have been attempted to reduce the skull’s high resistivity to TTFields, which increases the focal electric dose in the tumor and promotes prolonged survival [32,33]. TTFields treatment achieves the best efficacy at frequencies between 100 kHz and 400 kHz when used to treat GBMs. Whereas in clinical practice, TTFields is used at the intermediate frequency of 200 kHz in adults with GBMs, preclinical studies show that different GBM cell lines have different optimal electric frequencies, with that for KNS42 and GIN-31 cells being 200 kHz, that for SF188 being 400 kHz, and that for U87 being 100 kHz [34,35,36]. This phenomenon highlights the need for further investigation of the optimal frequency for patients with different tumor biofeatures to fulfill an individualized “TTFields prescription”, which could possibly be based on in vitro analysis of surgical specimens.

### 2.2. Biological Effects of TTFields on GBM Cells 

#### 2.2.1. Mitosis and Cell Cycle

The major therapeutic mechanism of action underlying the effects of TTFields used at a low intensity (1–2 V/cm) and medium frequency (100–400 kHz) is thought to occur through decreasing tumor cell proliferation and subsequently promoting cell death by altering the cell cycle of cells in mitosis (Figure 2). The use of TTFields is mainly effective on highly proliferating cancerous cells but has little impact on nonproliferating cells [25]. The primary mechanisms of action by which TTFields disrupts the cell cycle and mitosis include the following:

(1) Macromolecules and organelles that are responsible for mitosis are highly polar. Under the action of electric field forces, microtubulin dimers in cancer cells align with the direction of the electric field, which interferes with normal microtubule polymerization–depolymerization and results in abnormal spindle formation and subsequently prolonged mitosis and cell senescence [10,25,37,38].

(2) TTFields perturbs the localization of septins during anaphase, disrupts cell division, and leads to mitotic catastrophe [39].

(3) At the end of mitosis, when the cell is constricted into two daughter cells, it exhibits an “hourglass” structure and unevenly distributes the electric field across the dividing cell. In this context, macromolecules and organelles are pulled toward the cleavage furrow by electrophoretic forces, causing chromosomal mis-segregation during telophase and producing abnormal dividing cells with uneven numbers of chromosomes, which leads to cell membrane rupture and consequent cell death [37,40,41].

#### 2.2.2. Cell Autophagy

Autophagy is a mechanism that suppresses tumor growth during the early stage of tumorigenesis [42]. Aneuploidy formed in daughter cells caused by TTFields therapy is associated with the induction of the activity of regulators of autophagy and lysosomal gene expression [43,44]. Autophagy is a critical pathway of cell death in response to aberrant mitosis triggered by TTFields that has been observed to occur in several cancer cell lines [11,45]. Autophagy in response to TTFields therapy in different cell lines may present in different manners [46]. Autophagy-related morphological presentations include loss of plasma membrane integrity, lysosome accumulation, apparent vacuole formation, the appearance of a double-membraned autophagosome, and increased outflow of adenosine triphosphate.

#### 2.2.3. DNA Damage Repair and Replication

TTFields also slows tumor proliferation by delaying DNA repair and enhancing stress during DNA replication. The breast cancer susceptibility genes BRCA1 and BRCA2 function in the DNA damage response by mediating homologous recombination during the S and G2 phases of the cell cycle to maintain replication fidelity [47]. The BRCA DNA-damage response was found to be significantly inhibited during TTFields treatment, and the levels of foci of DNA double-strand break repair and chromatid-type aberrations were significantly increased within the cells exposed to TTFields [48]. In addition to disturbing DNA damage repair, TTFields also induces stress during DNA replication, causing decreased replication fork speed, increased replication errors, the development of R-loops with three-stranded nucleic acid structures, and single- or double-strand breaks [49].

Disturbing the DNA repair process is the key mechanism underlying the use of combination therapy with TTFields and other treatment modalities. Recent studies have shown that TTFields therapy can synergistically enhance the antitumor effects of radiotherapy and chemotherapy, possibly by blocking homologous recombination repair in irradiated tumor cells harboring irradiation- or chemically induced DNA damage [46,48,50]. Recent findings in multiple cancer types suggest the use of TTFields as a synergistic therapy with radiation or DNA-damaging drugs to promote the apoptosis of cancer cells [50,51,52,53,54]. Furthermore, TTFields can be used to overcome multiple-drug-resistant cancer cells with ABC transporter overexpression [55] and even sensitize targeted therapy for human epidermal growth factor receptor 2 in trastuzumab-resistant breast cancers [56]. These findings encourage the increased use of TTFields in combination with additional clinical treatment methods—the so-called cancer cocktail therapy—to maximize the antitumor benefit.

#### 2.2.4. Cell Migration and Metastasis

One of the key reasons why GBMs cannot be totally resected is their highly invasive and metastatic characteristics, which is one of the most crucial cancer hallmarks [57,58]. Several studies using various tumor models have proven that TTFields can extend survival by inhibiting tumor metastatic spread, seeding, and growth by preventing angiogenesis and downregulating the expression of epithelial–mesenchymal transition-associated proteins, including actin, vimentin, and cadherin [12,59,60]. TTFields also interferes with the directionality of cancer migration by inducing changes in the organization and dynamics of microtubules and actin [7]. Additionally, TTFields downregulates the expression levels of VEGF, HIF1-α, MMP2, and MMP9, which are the basis of tumor growth, invasion, metastasis, and recidivism, respectively [12]. Cilia are present in more than 30% of glioma cells and play a role in promoting cancer growth, migration, differentiation, and TMZ chemoresistance [14,61,62]. Shi et al. [15] found that TTFields exerted suppressing effects on primary cilia in both low- and high-grade glioma cell lines but fewer effects on normal astrocytes and neurons.

#### 2.2.5. Cell Metabolism

Another hallmark of cancer is the dramatically increased consumption of nutrients due to the reprogramming of cellular metabolism that occurs to support uncontrolled growth [63]. Even under conditions of abundant oxygen, cancer cells exhibit increased levels of glucose consumption and produce more lactate than normal cells [64]. TTFields has been found to inhibit the expression of pyruvate kinase M2, which is associated with elevated uptake of glucose, increased production of lactate, and reduced oxygen consumption, leading to reduced cell metabolism [65,66].

#### 2.2.6. Integrity of the Cell Membrane and the Blood–Brain Barrier (BBB)

Although the intensity of the electric field used in TTFields is only one hundredth of what is usually applied for electroporation, the use of TTFields can damage the membrane integrity of cancer cells by causing nanometer-sized holes [67], making it possible for GBM cells to be more permeable to particles with sizes of 4 kDa to 20 kDa. In this way, TTFields can increase the levels of 5-aminolevulinic acid uptake by GBM cells and assist in delineating the tumor–brain border during tumor resection [68,69,70]. This effect was not observed in normal fibroblasts and was reversible after 24 h. Moreover, the application of TTFields also disrupts the integrity of the BBB, removing obstacles of chemotherapeutic drug delivery into the tumor core [71,72]. During TTFields treatment, the uptake levels of several chemotherapeutic drugs was reported to be increased, which could be another mechanism underlying the synergistic effect of TTFields with chemotherapy [73]. TTFields activates calcium channels on the cellular membrane, interfering with the electrosignaling of glioma cells [74], which has been reported to promote glioma formation [75]. These effects of TTFields in elevating membrane and BBB permeability suggest a promising new method for application in the treatment of GBMs; this approach could be used to help deliver nonpermeable pharmacological agents to tumors and potentially promote antitumor outcomes.

### 2.3. Biological Effects of TTFields on the Tumor Microenvironment (TME)

In addition to the direct inhibitory effects on GBM cells, TTFields also changes the TME of GBM cells, especially the immune TME, to regulate tumor progression in an indirect manner. GBM cells are surrounded by a profoundly immunosuppressive, or immune-cold, environment [76]. In contrast to TMZ and radiation, which induce strong immune suppression, the use of TTFields therapy was demonstrated to activate the immune TME. Chen et al. [13] showed that the use of TTFields could lead to the cytosolic release of large micronucleus clusters through focal nuclear envelope disruption in GBM cells. These naked micronucleus clusters in the cytoplasm could subsequently recruit DNA sensors, including cyclic GMP-AMP synthase (cGAS) and absent in melanoma 2 (AIM2), activating their corresponding cGAS/STING and AIM2/caspase-1 inflammasomes and upregulating the expression levels of proinflammatory cytokines, type-1 interferons (T1IFNs), and T1IFN-responsive genes. As a result, the number of infiltrating activated dendritic cells (DCs), macrophages, and T cells increases, turning the “cold” TME of GBM into a “hot” TME and generating effective antitumor immunity against GBM cells. Moreover, adaptive immunity, as shown by studies on peripheral blood mononuclear cells (PBMCs), can also be activated after TTFields treatment [13]. The use of TTFields recruits DCs from bone marrow, promotes engulfment of cancer cells by bone-marrow-derived DCs (BMDCs), and assists in DC maturation. TTFields-treated cells can in turn promote DC maturation by upregulating the expression levels of MHC class II, CD40, and CD80 when cocultured with BMDCs [68]. The combination of the use of TTFields and anti-PD-1 therapies leads to a significant decline in tumor volume and remarkably higher proportions of tumor-infiltrating T cells, macrophages, DCs, and antitumor cytokines than the use of monotherapy [8]. TTFields also induces the production of nitric oxide and reactive oxygen species and elevates the expression levels of proinflammatory cytokines in macrophages via the MAPK and NF-kB signaling pathways [77].

Unlike monocytes/macrophages, activated T cells undergo rapid expansion to generate subclones, which are vital to the specific immune response to cancer cells. An in vitro experiment revealed the inhibitory effect of TTFields on proliferating T cells [78], although no evidence of T-cell reduction was found in various clinical studies. Notwithstanding the inhibition of T-cell proliferation, TTFields has no significant effect on the overall functionality of T cells, which exhibit preservations in the secretion of IFN-γ and cytotoxic degranulation, rendering it possible for use in combination with chimeric antigen receptor T-cell (CAR-T) immunotherapy [78,79].

Moreover, the use of TTFields temporally reduces microvascular density to suppress tumor growth by impairing the integrity of the BBB [80]. These characteristics could facilitate the infiltration of immune cells and eventually modify the immune-suppressive TME of GBMs toward an immune-activating TME.

## 3. Clinical Studies of TTFields Treatment on GBMs

### 3.1. TTFields Apparatus Applied in the Clinic

Novocure’s Optune^®^ is the most widely used electric field therapy device worldwide, consisting of an electrical field generator, two pairs of scalp-adhesive transducer arrays, a messenger bag, connection cables, portable batteries and chargers, and a power supply (Figure 1d). Weighing just 2.7 pounds, Optune^®^ is easily wearable and portable, enabling carrying comfort and continuous treatment almost anywhere and anytime. The second generation of the Optune^®^ system is ergonomically improved relative to the first-generation device, with a significantly smaller size and lower weight (Figure 1e). In the field of GBM therapy, Optune^®^ is designed to treat adult patients aged 22 years or older. The treatment-planning software NovoTAL, which uses computer-generated algorithms, optimizes the electric field intensity and array location based on magnetic resonance imaging from patients to enable field emanation through the scalp and skull to the tumor [81].

Optune^®^ is currently available in many countries, including the United States, Europe, Japan, and China. More than 18,000 patients have started therapy with the device (https://www.optune.com/ accessed on: 1 June 2022). However, the unbalanced distribution of devices is still an issue, as only two-fifths of surveyed centers worldwide had TTFields available to offer to GBM patients [82]. Similar electric field therapy devices are in the process of development. For example, in Japan, the Electro-Capacitive Cancer Therapy device, developed by Dr. Warsito P. Taruno at Shizuoka University in collaboration with CTech Labs Edward Technology Company, has been approved for use by the Regenerative Medicine Act (Figure 1f). In China, the EFE-G100 device was developed by Jiangsu Hailai Xinchuang Medical Technology Co. (Nanjing, China).

### 3.2. Initial Trials of TTFields Applied in Human GBM Patients

The first trial of TTFields treatment in human GBMs was conducted in 2002 (EF-02) as a pilot study using the NovoTTF-100A™ instrument in six patients with advanced malignant tumors, including one with melanoma, one with pleural mesothelioma, one with GBM, and three with breast cancer [83]. Unfortunately, the patient with TMZ- and carmustine-resistant GBM showed no response to TTFields treatment, possibly due to the short treatment duration of only 4 weeks. However, this study confirmed the safety profile of the use of TTFields, with a high patient compliance of > 80%, implying the potential of TTFields as a new treatment option for refractory, advanced tumors (Table 1).

Kirson et al. [26] conducted the second landmark trial of TTFields (EF-07) on 10 patients with rGBMs (Table 1). In this study, with prolonged used of TTFields, the median time to disease progression was 26.1 weeks, and the median OS was 62.2 weeks, which was more than double the medians observed in historical controls. A case report later showed that two patients with rGBMs were still alive in 2012 [84]. In 2009, a second group of 10 ndGBM patients was included after success in the treatment of rGBMs (Table 1). Kirson et al. [85] reported that patients with ndGBMs who were treated with TTFields plus maintenance TMZ therapy after radiotherapy had a longer median OS of more than 39 months and a longer median PFS of 155 weeks than the OS of 14.7 months and PFS of 31 weeks observed in matched historical controls receiving maintenance TMZ alone. These studies set the foundation for subsequent large-scale randomized, controlled trials involving the application of TTFields in patients with rGBMs and ndGBMs.

**Table 1 cancers-14-03669-t001:** Landmark clinical trials of TTFields in treating glioblastoma.

Study	Year	Phase	Arms	Patients	Tumor Type	Treatment Protocol	mOS	mPFS	Systemic AEs	Skin Toxicity
EF-14	2017 [4]	3	2	695	ndGBM	Arm1 (*n* = 466): TTFields plus maintenance temozolomide chemotherapy after tumor resection or biopsy and concomitant radiochemotherapy	20.9 months	6.7 months	48%	52%
						Arm2 (*n* = 229): temozolomide alone after tumor resection or biopsy and concomitant radiochemotherapy	16.0 months	4.0 months	44%	0%
EF-14	2015 [5]	3	2	315	ndGBM	Arm1 (*n* = 210): TTFields plus maintenance temozolomide chemotherapy after tumor resection or biopsy and concomitant radiochemotherapy	20.5 months	7.1 months	44%	43%
						Arm2 (*n* = 105): temozolomide alone after tumor resection or biopsy and concomitant radiochemotherapy	15.6 months	4.0 months	44%	0%
EF-11	2012 [16]	3	2	237	rGBM	Arm1 (*n* = 120): TTFields alone	6.6 months	2.2 months	0%	16%
						Arm2 (*n* = 117): chemotherapy (physician’s best choice)	6.0 months	2.1 months	16%	0%
EF-07	2009 [85]	1	1	10	ndGBM	TTFields combined with maintenance temozolomide after surgery and radiation therapy	>39 months	155 weeks	0%	100%
EF-07	2007 [26]	1	1	10	rGBM	Continuous TTFields after adjuvant temozolomide and brain surgery and/or radiotherapy for the primary tumor	62.2 weeks	26.1 weeks	0%	90%
EF-02	2008 [83]	1	1	1	rGBM	Continuous TTFields treatment for at least 4 weeks after heavily pretreatment with several lines of therapy	Not available	Not available	0%	Not available

### 3.3. Clinical Efficacy of TTFields in rGBM Patients

To date, there is no standard treatment for rGBMs. Before the introduction of TTFields, clinical trials, reoperation, chemotherapy, radiation, targeted therapy, and immunotherapy were potential treatment options. Among them, bevacizumab, a vascular endothelial growth factor (VEGF) inhibitor, is the most promising treatment choice. However, bevacizumab was shown to only provide benefits in PFS, with no significant change in patient OS [86].

The use of TTFields was approved by the FDA for rGBM treatment in 2011 and was included in the NCCN guidelines in 2013 because of its promising efficacy demonstrated in the EF-11 trial [16]. This phase 3 controlled trial demonstrated the efficacy and safety of TTFields in treating rGBMs (Table 1). A total of 237 patients were included in the study, among whom 120 were randomized to be treated with TTFields alone (>18 h/d), whereas the others were treated with PBC therapy. The median OS was 6.6 months and 6.0 months (*p* = 0.27) in the TTFields and PBC groups, respectively, and the 6-month PFS was 21.4% and 15.1% (*p* = 0.13), respectively. Although rGBM patient survival was not better with the use of TTFields than with the use of PBC, this chemotherapy-free treatment had effects that appeared to be comparable to those of chemotherapy; most importantly, it induced less toxicity and improved QoL [16].

### 3.4. Clinical Efficacy of TTFields in ndGBMs

For ndGBMs, the standard of care is maximal safe surgical removal, followed by radiation plus concurrent TMZ, as well as subsequent TMZ maintenance therapy. This standard-of-care Stupp protocol prolongs the OS from the 12.1 months achieved with postoperative radiation alone to 14.6 months in ndGBM patients [3].

The use of TTFields was approved by the FDA for ndGBM in 2015 and was included in the NCCN guidelines as a category 1 recommendation in 2018 because of its high clinical efficacy. In 2009, a phase 3 controlled trial (EF-14) was launched to test the efficacy and safety of TTFields in combination with TMZ maintenance therapy for ndGBM patients (Table 1). A total of 695 patients who had completed surgery and chemoradiotherapy were included. Two-thirds of the subjects were randomized to be treated with TTFields (>18 h/d) plus adjuvant TMZ, whereas the others were given standard adjuvant TMZ maintenance therapy. An interim analysis in 2015 reported that the median PFS of the TTFields plus TMZ group and TMZ-alone group was 7.1 months and 4.0 months, respectively, and the median OS was reported to be 20.5 months and 15.6 months [5]. The final report published in 2017 demonstrated that the addition of TTFields to TMZ maintenance therapy after chemoradiotherapy increased patient OS from the 16.0 months achieved using TMZ therapy alone to 20.9 months and the PFS from 4.0 months to 6.7 months [4]. Subgroup analyses of the EF-14 trial showed that increased compliance with TTFields therapy was an independent prognostic factor for improved patient survival. For patients using TTFields > 22 h each day, the 5-year survival rate was high, reaching 29.3% [87].

### 3.5. Combination Therapy with TTFields

#### 3.5.1. TTFields Combined with Chemotherapy

GBMs develop chemoresistance due to various mechanisms, including activated DNA repair, angiogenesis, hypoxic TME and acidosis, immune escape, and GBM stem cell development [88]. Moreover, the BBB, a major hurdle for the efficient delivery of chemotherapy agents, also contributes to GBM chemoresistance [89]. Identifying ways to improve chemoresistance has become an urgent issue. Prior studies on non-small-cell lung cancer demonstrated that the use of TTFields improved the treatment efficacy when combined with pemetrexed, cisplatin, paclitaxel, erlotinib, TMZ, and 5-FU [53]. Strategies to improve therapeutic outcomes in GBM patients by combining TTFields with TMZ therapy have been extensively studied. Preclinical data showed that the use of TTFields and alkylation agents led to additive or synergistic effects on GBM patients, and TMZ-resistant glioma cells responded well to TTFields treatment, highlighting the clinical potential of this combination treatment approach [27]. Kirson et al. [85] showed that the use of TTFields can increase the sensitivity of GBM cells to TMZ, making it possible to achieve similar or even improved therapeutic effects with lower dosages, thus reducing the overall toxicity. Moreover, a pilot clinical study (EF-07) reported a significantly improved therapeutic effect in those using TTFields/TMZ combined therapy than in those using maintenance TMZ alone, which further corroborated the authors’ expectation [85]. The final result of the EF-14 trial in 2017 also showed that the use of the combination treatment with TTFields and TMZ resulted in significantly higher PFS and OS than the use of TMZ maintenance therapy alone [4]. Subsequently, researchers from South Korea performed a subgroup analysis of 39 patients in the EF-14 trial, showing that the median PFS was 6.2 months in the combination treatment group and 4.2 months in the group treated with TMZ alone; the median OS was 27.2 months in the combination treatment group and 15.2 months in the group treated with TMZ alone, similar to the overall results observed in the EF-14 trial [90].

In addition to the use of TMZ, the use of combination treatments with TTFields and other chemotherapeutic agents showed clinical efficacy. Preclinical studies have shown that TTFields and withaferin A synergistically inhibit the proliferation of GBM2/GBM39/U87-MG cells [73]. The NOA-09/CeTeG trial found that the combination of lomustine and TMZ was superior to TMZ monotherapy in patients with O6-methylguanine DNA methyltransferase (MGMT) promoter methylation (mMGMT) ndGBMs [91]. In 2020, Lazaridis et al. [52] reported the results of a retrospective analysis of mMGMT ndGBM patients receiving TTFields in combination with lomustine and TMZ, with a median PFS of 20 months, revealing a potential clinical benefit.

#### 3.5.2. TTFields Combined with Radiotherapy

TTFields therapy synergistically enhances the efficacy of radiation in glioma cells [46]. Preclinical evidence suggests that the combination of radiation and TTFields therapy prevents GBM cells from migrating and invading and promotes cell apoptosis, DNA damage, and mitotic abnormalities [92,93]. In 2020, a study with the aim of examining the safety and efficacy of TTFields in combination with TMZ and radiotherapy was reported [94]. A total of 10 patients with ndGBM received TTFields/radiation/TMZ followed by adjuvant TMZ/TTFields, achieving a median PFS of 8.9 months from enrollment. In addition, Stein et al. [95] reported a case of thalamic GBM, IDH wild-type, showing a complete radiological response after chemoradiation with TMZ, proton boost therapy, and TMZ maintenance in combination with TTFields therapy. Recently, Miller et al. [96] evaluated the skin toxicity of scalp-sparing chemoradiation plus TTFields followed by maintenance TMZ plus TTFields in 30 patients with ndGBMs, showing good tolerance of the new protocol with no need to remove electric arrays during the radiation process, as well as a higher PFS in these patients than in the historical controls.

#### 3.5.3. TTFields Combined with Targeted Therapy

The use of TTFields combined with the VEGF inhibitor bevacizumab in the treatment of GBMs has attracted considerable attention, and many phase 2 trials are being conducted. One such trial was a retrospective study of 48 patients with rGBMs. The two cohorts received TMZ, bevacizumab, irinotecan, and TTFields (TBI + T) or bevacizumab-based chemotherapy with TTFields. The median OS and PFS for patients treated with TBI + T were 18.9 months and 10.7 months, respectively, compared with 11.8 months and 4.7 months in the bevacizumab group [97]. Another study divided patients with rGBMs into two groups: patients treated with NovoTTF-100ATM and bevacizumab and patients treated with NovoTTF-100ATM, bevacizumab, 6-thioguine, lomustine, capecitabine, and celecoxib (TCCC). The results showed that tumors were smaller in patients treated with NovoTTF-100ATM, bevacizumab, and TCCC. Although the compliance of the cohort receiving NovoTTF-100ATM, bevacizumab, and TCCC was poor, they exhibited a longer median OS (10.3 vs. 4.1 months) and a longer median PFS (8.1 vs. 2.8 months) [98]. Elzinga and Wong [99] reported that the addition of TTFields therapy led to resolution of the recurrent cystic GBM, as well as most of the surrounding cerebral edema, in a patient with an unfavorable response to bevacizumab. Ansstas and Tran [100] reported that eight patients with rGBMs who exhibited disease progression on bevacizumab underwent treatment with TTFields alone. Following TTFields therapy, the median patient OS from the last dose of bevacizumab was approximately 8 months, which was almost twice that in historical controls with bevacizumab failures.

Other targeted agents combined with TTFields therapy have also been explored. For instance, Meletath et al. [101] reported a case in which TTFields was used in combination with dabrafenib, an inhibitor of BRAFV600E, and produced a significant clinical and radiological response in patients with advanced gliomas with BRAFV600E mutations. Kim et al. [102] confirmed that sorafenib combined with the use of TTFields improved the treatment outcome of GBMs by downregulating STAT3 expression levels in vivo and in vitro. Kessler et al. [103] demonstrated that spindle assembly checkpoint inhibition augmented the effect of TTFields on U-87MG and GaMG cells.

#### 3.5.4. TTFields Combined with Immunotherapy

Recently, immunotherapy has become a hot spot and forefront of research with its success in treating many solid and blood cancers. Various immunotherapies have been investigated to treat GBMs, and several clinical trials have been conducted, including those for checkpoint inhibitors, vaccines, adoptive lymphocyte transfer, and oncolytic therapy, although with few encouraging findings [104]. Although no clinical trials have been published involving the use of immunotherapy in combination with TTFields, it cannot be denied that this new method may produce some breakthroughs, considering the effect of TTFields on the immune TME [8,13,77], which is promising.

#### 3.5.5. TTFields Combined with Other Treatment Modalities

The skull is one of the layers between electric arrays and the tumor bed that presents the most prominent attenuation of the electric intensity of TTFields [105]. Korshoej et al. [32] reported a trial testing the combination of skull remodeling surgery (SR surgery) with TTFields in patients with rGBMs of first relapse. SR surgery was performed by drilling five 15 mm diameter holes above the tumor resection cavity to reduce the resistance in TTFields. This phase 1 trial (NCT02893137) showed that the combination of SR surgery and TTFields treatment was safe and feasible and improved patient OS, with a median OS of 15.5 months and a median PFS of 4.6 months. On this basis, the OptimalTTF-2 phase 2 trial (NCT0422399) was launched in November 2020 and is currently ongoing. Jo et al. [106] evaluated the effects of combining the use of hyperthermia and TTFields on GBM cells, demonstrating that combined therapy induced inhibition of cell migration, higher apoptosis rates, and increased downregulation of STAT3 expression levels than the use of hyperthermia or TTFields alone.

#### 3.5.6. Use of TTFields in Pediatric GBM Patients

Fewer studies have been conducted using TTFields to treat pediatric GBM patients than adult GBM patients. Green et al. [107] reported the use of TTFields and chemotherapy and/or radiotherapy in pediatric patients with high-grade gliomas, showing that all patients tolerated TTFields well. Recently, Gott et al. [108] reported that the use of TTFields in a 3-year-old patient with H3K27 M-mutated diffuse midline glioma was feasible and safe.

### 3.6. Identification of Distinct Response to TTFields Treatment

Studies were conducted to identify predictive biomarkers of the efficacy of the use of TTFields in GBM patients. A retrospective review of 149 patients with IDH wild-type rGBMs, of whom 29 were treated with TTFields, found that PTEN mutation might predict prolonged postprogression survival better in the TTFields-treated group than in the groups subjected to other treatments, whereas patients with PTEN wild-type rGBMs showed no improvements [109]. A recent genomic analysis revealed that molecular driver alterations in NF1, as well as wild-type PIK3CA and EGFR, were associated with improved response to TTFields [110]. Radiological examinations were also applied to detect treatment response to TTFields as early as 2–3 months after the start of TTFields treatment, and the findings included metabolic change of the reduction in the choline/creatine ratio in ndGBMs using physiologic and metabolic MRI [111] and a decrease in tryptophan uptake in rGBMs based on amino acid PET scanning with alpha[C-11]-methyl-L-tryptophan [112], although more clinical studies are required for these potential applications in the future.

### 3.7. Safety/Adverse Events

The use of TTFields promotes improved clinical outcomes and exhibits no known systemic toxicity. The most predominant local adverse events (AEs) associated with the use of TTFields treatment for GBMs are dermatologic events due to the continuous contact between the arrays and the shaved scalp. TTFields-associated skin reactions include allergic or irritant dermatitis; xerosis or pruritus; mechanical lesions; hyperhidrosis; and, more rarely, skin erosion, infections, and ulcers [113,114,115]. The causes of dermatologic AEs are diverse, including a moist occluded scalp environment, chronic use of steroidal medicine and systemic anticancer drugs, and irritation of the skin at the site of the previous surgical wound by the liquid medium of the electrode array [116,117,118,119,120,121].

Because survival benefits positively correlate with the continuity of TTFields treatment [87], continuous use is highly recommended, and skin events are somewhat inevitable. Concerns regarding skin reactions should not be a barrier to continuing TTFields therapy, as most of the dermatologic AEs are mild to moderate (grade 1/2), while very few patients (only 2% in EF-14) experience severe skin involvement (≥grade 3 AE) [4,114].

Although TTFields therapy results in dermatologic AEs in a large number of patients with GBMs, dermatologic AEs are mostly reversible and manageable [16]. Prophylactic interventions, in combination with early identification and prompt topical therapies, help maintain improved skin conditions, supporting patient compliance with continuous TTFields therapy. Recommendations for preventing TTFields-associated dermatologic AEs include patient and family education, proper shaving to avoid cuts, cleaning and drying of the scalp, prevention of skin infection, scar reduction, and timely array repositioning [114,120]. An increase in scalp dose was detected when patients were treated with radiation and concurrent TTFields, and a scalp-sparing protocol could optimally mitigate skin toxicity [122].

### 3.8. Health-Related Quality of Life

It is crucial to address the effect of TTFields treatment on patient well-being, as reflected by health-related QoL (HRQoL), in addition to the prolongation of life. As reported in EF-11, there were no differences observed in global health and social functioning domains between TTFields treatment and chemotherapy groups, as assessed using the EORTC QLQ-C30 questionnaire. The scores of cognitive, emotional, and role functioning were higher, whereas physical functioning was slightly worse in the TTFields group [16]. In EF-14, no significant differences were detected between the TTFields plus maintenance TMZ group and the group treated with TMZ alone with respect to HRQoL, except that more incidences of itchy skin were observed in the TTFields group [123]. Recently, a large-scale, real-world study of HRQoL in GBM patients using TTFields revealed that a longer duration of TTFields use was strongly associated with improved HRQoL, especially in progressed patients [124]. Because patients need to continuously carry the electric device, remain alopecic, and avoid wearing wigs, TTFields-related negative impacts on patient QoL, apart from the health-related aspects, also need to be investigated [125].

### 3.9. Real-World Cost-Effectiveness

Although TTFields technology is evolving and discount options are provided, it remains an extremely high-cost treatment, with prices that are far higher than those of the conventional treatment modalities for GBMs. Studies from France showed that the incremental cost-effectiveness ratio (ICER) of TTFields is at approximately EUR 510,273 to EUR 549,909 per life year gained, which is largely outside the widely recognized willingness-to-pay thresholds [126,127], unlike the ICER of TMZ-assisted radiotherapy, at approximately USD 55,000 per life year gained [128]. However, researchers from the United States demonstrated that this value for TTFields was only USD 150,452 per life year gained, which is within the willingness-to-pay thresholds [129]. Because the existing results are conflicting, future studies concerning the cost-effectiveness of TTFields are still needed to acquire a more accurate assessment in real-world settings. Substantial price regulation by health administrations is urgent and may assist in making this promising therapy more affordable and accessible to GBM patients, especially in developing and less developed countries. It is also important to maintain incentives for innovation while managing product prices.

## 4. Ongoing Trials of TTFields Use in GBM Patients

The milestone clinical trials EF-11 and EF-14 laid the foundation for the use of TTFields as the standard of care in GBM patients. However, the optimal starting time of TTFields for ndGBM patients, clinical efficacy as a concurrent treatment modality for rGBM patients, detailed mechanism of action, radiological and pathological signatures after TTFields treatment, and possible ways to improve patient compliance are still under investigation. There are many ongoing trials aiming to settle these currently unsolved clinical questions (Table 2).

In the NCT03258021 trial (TIGER) being conducted in Germany, researchers are attempting to include 710 patients with ndGBMs with clinical indication and set the use of TTFields as a routine clinical practice. Apart from OS, PFS, and serious AEs, researchers are aiming to collect data on the timing of the use of TTFields and reasons that patients are refusing, which may provide a possible reference for device promotion and elevate patient compliance. As a major follow-up study to the EF-11 [16] and EF-14 [4,5] trials, the aim of the EF-32 (NCT04471844) trial is to enroll a total of 950 ndGBM patients and assess whether the earlier application of TTFields at the time of concurrent chemoradiation improves patient survival more than the standard of care [4]. Similarly, in the NCT03705351 pilot trial, eligible patients are being recruited and receiving TTFields therapy starting < 2 weeks prior to the start of concurrent chemoradiation; the aim of the researchers is to assess the incidence rate and severity of AEs associated with trimodal therapy. Trial EF-33 (NCT04492163) is recruiting 25 patients with rGBMs and arranging for them to receive continuous TTFields treatment with high-intensity transducer arrays to assess its benefits in terms of patient survival.

Some ongoing trials are further exploring the mechanisms of action underlying the effects of TTFields. For instance, in trial NCT03194971, pathological information on tumor cellularity, apoptosis status, and tumor cell histomorphometry at biopsy of patients with either ndGBMs or rGBMs will be collected to identify pathological signatures and patterns of failure after TTFields treatment. In addition, preclinical studies have demonstrated that TTFields therapy enhances the damaging effects of radiotherapy in U118 cells by inhibiting DNA repair [46]. Ongoing trials are seeking to explore the therapeutic effect of combination treatment with TTFields and radiotherapy in rGBMs. For example, trial NCT04671459 is expected to evaluate the use of TTFields plus radiosurgery plus/minus FET-PET imaging to define tumor volume in 40 patients with rGBMs.

In addition, several trials have used MRI to examine, from a radiological angle, how TTFields therapy affects GBMs (NCT02441322, NCT03297125, and NCT03642080). These trials are intend to optimize treatment regimens and to establish reliable assessments to predict which groups of patients are most likely to achieve clinical benefits with the use of TTFields. Moreover, ongoing trials are exploring the clinical effects of the use of TTFields in patients receiving chemotherapy agents other than TMZ. Because TTFields can reduce DNA double-strand repair by downregulating the activity of the BRCA1 signaling pathway, tumor cells may be more sensitive to the blockade of DNA repair caused by poly-ADP ribose polymerase (PARP) inhibition [130]. Therefore, in trial NCT04221503, TTFields and niraparib, a PARP inhibitor, will be used to treat rGBMs. Selinexor is a nuclear export-selective inhibitor for multiple myeloma [131]. In trial NCT04421378, there was also a group of patients being treated with TTFields and combined selinexor.

The increasing use of immunotherapy provides a direction for its combination with TTFields. The aim of the phase 1 trial NCT03223103 is to test the tolerability and safety of a mutation-derived tumor antigen vaccine combined with the use of TTFields in the maintenance phase of TMZ therapy in patients with ndGBMs. The phase 2 trial NCT03405792 (2-THE-TOP) investigates whether pembrolizumab, an anti-PD-1 monoclonal antibody, enhances TTFields-induced GBM-specific immune responses in ndGBM patients. The preliminary result showed an improved PFS of 12.1 months and an OS of 25.2 months compared with 7.9 months and 15.9 months, respectively, for matched control patients in the EF-14 trial (https://www.novocure.com/updated-2-the-top-data-suggest-improvements-in-progression-free-survival-overall-survival-compared-to-matched-control-patients-from-ef-14-trial/ accessed on: 1 June 2022). This exciting result further extends our understanding of the mechanisms by which TTFields enhances the immune TME of GBMs [13,132].

Given that TTFields is a new treatment method, its impacts on patient QoL in all domains deserve further attention. In many ongoing clinical trials, QoL and safety/AEs are important endpoints (NCT03258021, NCT04421378, NCT04218019, NCT03223103, NCT04469075, NCT04474353, NCT03705351, and NCT04397679).

## 5. Conclusions and Perspectives

TTFields therapy, a noninvasive and innovative therapeutic approach, has emerged as the fourth most effective treatment option for the management of GBMs in humans. According to milestone large-scale phase 3 randomized, controlled trials and the following real-world data, TTFields therapy leads to substantially prolonged patient survival and acceptable and reversible mild-to-moderate AEs (Table 3).

Clinical trials are ongoing to further evaluate the safety and efficacy of TTFields in treating deadly tumors, as well as its biological and radiological influences. Major mechanisms of action underlying the effects of TTFields therapy performed by delivering low-intensity, intermediate-frequency, alternating electric fields to GBMs include disruptions in cancer cell mitosis, delays in the DNA-damage-repair process, enhancements in autophagy, and inhibition of tumor cell metabolism and tumor cell migration. The ability to use of TTFields to strengthen intratumoral immunity, increase the permeability of the tumor membrane and the BBB, and disrupt the repair process of radiation- or chemotherapy-induced DNA damage makes it a promising synergistic therapy for use with the existing standard-of-care treatment protocol for GBMs. However, the overall acceptance of the use of TTFields in the real world remains at a low level. One of the main reasons is the imperfectly elucidated mechanism of action among neurosurgeons and neuro-oncologists, leading to reluctance to recommend TTFields to patients with GBMs. Others include the high treatment cost beyond the willingness-to-pay threshold and severely low social acceptance, which is due to the persistence of visible sticky patches on the shaved head. Continuous research into the mechanisms of action, substantial price regulation, and development of skull-remodeling surgery or novel intracranial electrodes may assist in increasing the use and acceptance of TTFields among both healthcare workers and patients and in turn improve the prognosis of the deadliest brain malignancy.

## Figures and Tables

**Figure 1 cancers-14-03669-f001:**
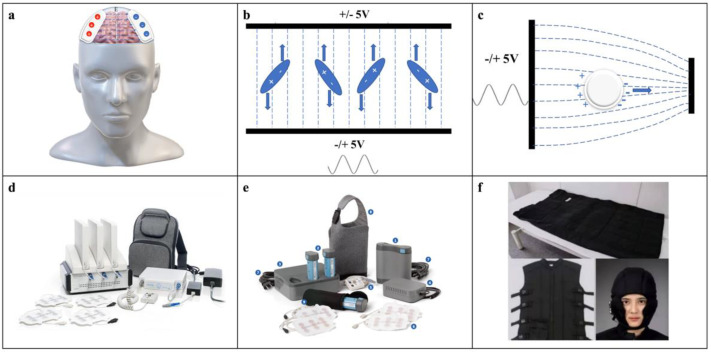
Electromagnetism of TTFields and apparatus used in clinical practice. (**a**) Model of the TTFields device for GBM treatment. (**b**) Electropolar molecules tend to be parallel to the direction of the electric field in an alternating electric field. (**c**) Movement of macromolecules in an inhomogeneous electric field resulting from the dielectrophoretic effect. (**d**) The first-generation Novocure Optune^®^ system. (**e**) The second-generation Novocure Optune^®^ system. (**d**,**e**) Reproduced with permission from Novocure GmbH © 2021 Novocure GmbH—All rights reserved. (**f**) The Electro-Capacitive Cancer Therapy device developed by Warsito at Shizuoka University in Japan; Used with permission.

**Figure 2 cancers-14-03669-f002:**
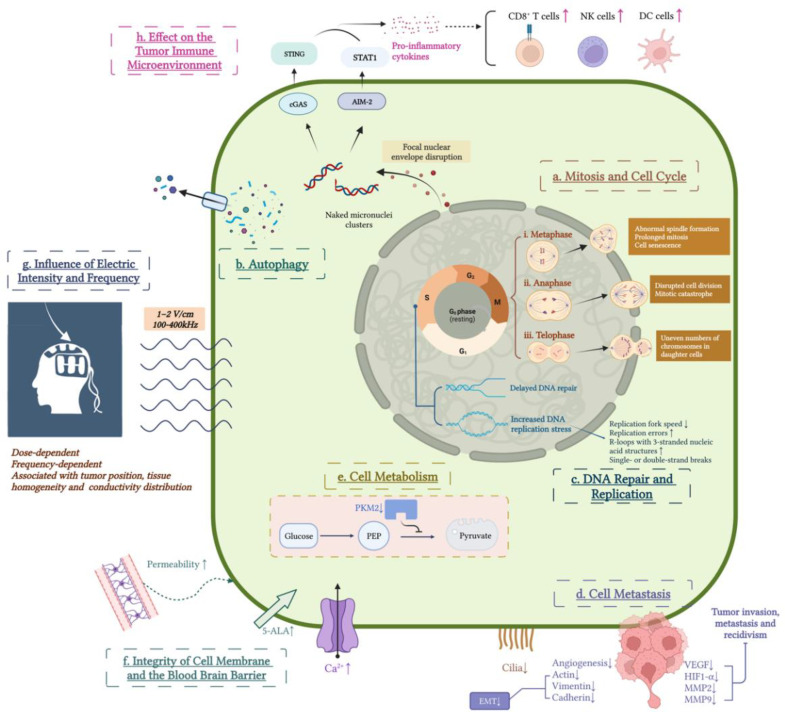
Biological mechanisms of action of TTFields in GBMs. (**a**) TTFields therapy disrupts the cell cycle and mitosis of GBMs in multiple phases, including (i) metaphase, (ii) anaphase, and (iii) telophase. (**b**) TTFields therapy enhances autophagy and leads to subsequent cell death by inducing aneuploidy in daughter cells. (**c**) TTFields therapy delays DNA repair and enhances DNA replication stress. (**d**) Cell metastasis is inhibited by TTFields through prevention of angiogenesis, downregulation of metastasis-related proteins, and suppression of the primary cilia. (**e**) TTFields therapy inhibits the expression of pyruvate kinase M2 and therefore reduces cell metabolism. (**f**) The integrity of the cell membrane and blood–brain barrier (BBB) is disrupted by TTFields, resulting in increased 5-aminolevulinic acid uptake, activated calcium channels on the membrane, and elevated transmission of nonpermeable pharmacological agents through the BBB to the tumors. (**g**) TTFields therapy shows diverse efficacy under different electric intensities and frequencies and is influenced by the conductivity of the skull, tumor position, and tissue homogeneity. (**h**) TTFields therapy changes the immune microenvironment of GBMs from ‘cold’ to ‘hot’ by upregulating proinflammatory cytokines and activating intratumoral infiltrated immune cells via the cGAS/STING and AIM2/caspase-1 pathways.

**Table 2 cancers-14-03669-t002:** Ongoing trials of TTFields in glioblastoma as of 15 May 2022.

Study Identifier	Status	Arms	Sample Size	Tumor Type	Intervention/Treatment	OS	PFS	AE	QoL	Others	Duration
NCT03501134 (ACTION)	Completed	1	20	ndGBM	TTFields				√	24w-MET-h/wk, 24w-sleep quality, 24w-mood state, 24w-functional capacity, 8/16/24w-average daily number of steps	3 years
NCT03033992	Recruiting	1	25	rGBM	TTFields			√		ORR, EFS	4 years
NCT03642080	Recruiting	1	48	ndGBM, rGBM	TTFields		√				5 years
NCT05086497	Not yet recruiting	2	155	ndGBM, rGBM	TTFields + conventional/advanced MR imaging array mapping layout	√	√				4 years
NCT05030298	Not yet recruiting	2	40	ndGBM	TTFields + RT + TMZ + radiosurgery	√	√	√		Toxicity	3 years
NCT02903069	Completed	Multi	66	ndGBM	MRZ + TMZ ± RT ± TTFields	√	√	√		MTD, drug serum concentrations	5 years
NCT04223999 (OptimalTTF-2)	Recruiting	2	70	rGBM	Skull-remodeling surgery ± TTFields	√	√	√	√	ORR, KPS	4 years
NCT04218019 (GERAS)	Suspended	2	68	ndGBM	Early/late TTFields		√	√	√	SCTR	2 years
NCT03223103	Active, not recruiting	1	13	ndGBM	Poly-ICLC + TTFields + peptides	√	√			DLT, ORR	5 years
NCT04469075	Recruiting	1	58	ndGBM	Clindamycin phosphate + triamcinolone acetonide				√	Grade 2 or higher skin toxicity	3 years
NCT04474353	Recruiting	1	12	ndGBM	TTFields + TMZ + SRS + gadolinium	√	√			DLT	3 years
NCT04689087	Recruiting	1	40	rGBM	TTFields + BPC	√	√	√			2 years
NCT04471844 (EF-32)	Recruiting	2	950	ndGBM	TTFields + TMZ + RT	√	√	√	√	ORR	6 years
NCT04221503	Recruiting	2	30	rGBM	Surgery + TTFields + niraparib	√	√	√		Disease control, ORR	6 years
NCT03258021 (TIGER)	Active, not recruiting	1	710	ndGBM	TTFields	√	√	√	√	Compliance, reason for refusing TTFields	4 years
NCT04671459 (TaRRGET)	Recruiting	1	40	rGBM	TTFields + SRS	√	√			Radiation necrosis range, steroid needs, ORR, patterns of failure	3 years
NCT02893137	Completed	1	15	ndGBM, rGBM	Craniectomy + TTFields	√	√	√	√	ORR	3 years
NCT04717739	Not yet recruiting	1	500	ndGBM, rGBM	TTFields			√	√	Compliance, sleep quality, neurocognitive functioning	2 years
NCT04421378	Recruiting	Multi	474	ndGBM, rGBM	Selinexor ± TTFields ± TMZ ± RT ± lomustine ± bevacizumab	√	√	√		Phase 1a: maximum tolerated dose, recommended phase 2 dose; Phase 1a/1b: TTP, drug serum concentrations; Phase 2: ORR	3 years
NCT04757662	Active, not recruiting	1	18	ndGBM	Tadalafil + TMZ + TTFields	√	√	√		MDSC change, severe lymphopenia, HDI	2 years
NCT00916409	Completed	2	700	ndGBM	TMZ ± TTFields	√	√				8 years
NCT04492163 (EF-33)	Recruiting	2	24	rGBM	TTFields	√	√	√		ORR	2 years
NCT01954576	Terminated	1	21	rGBM	TTFields		√		√	ORR, genetic signature of response	8 years
NCT03194971	Recruiting	2	20	ndGBM, rGBM	TTFields					States of mitotically cells	7 years
NCT03405792 (2-THE-TOP)	Active, not recruiting	1	31	ndGBM	TMZ + TTFields + pembrolizumab	√	√			Toxicity and tolerability, immune reaction by pembrolizumab	5 years

This table includes only ongoing trials for glioblastomas with a minimal sample size of 10. Abbreviations: ndGBM, newly diagnosed GBM; rGBM, recurrent GBM; TMZ, temozolomide; SRS, stereotactic radiosurgery; RT, radiotherapy; BPC, best physician’s choice; DLT, dose-limiting toxicities; OS, overall survival; AE, adverse event; Qol, quality of life; PFS, progression-free survival; MRZ, marizomib; MTD, maximum tolerated dose; SCTR, safely conducted therapy rate; ORR, overall response rate; EFS, event-free survival; HDI, heterogeneity diffusion imaging; MET-h/wk, mean change between baseline and week 24 in total physical activity; SCTR, safely conducted therapy rate; KPS, Karnofsky score; TTP, time to progression.

**Table 3 cancers-14-03669-t003:** Positive and negative characteristics of TTFields in treating glioblastomas.

Positive Characteristics	Negative Characteristics
**Mechanisms of action**	**Mechanisms of action**
Disturbing mitosis and cell cycle	Still imperfectly unelucidated
Delaying DNA damage repair process	
Enhancing cell autophagy	**Clinical efficacy**
Inhibiting cell metabolism and angiogenesis	Phase 3 trials on combination therapy are needed
Limiting cancer cell migration and metastasis	
Increasing the permeability of cancer cell membrane and blood–brain barrier	**Acceptance of the use of TTFields in the real world**
Strengthening intratumoral immunity by turning the “cold” TME into “hot”	Very low (<12% to <16%)
**Clinical efficacy**	
Prolonged OS and PFS in ndGBM patients	**Safety/adverse events**
Prolonged OS and PFS in rGBM patients	Dermatologic events, mostly mild to moderate
Additional survival benefit when combined with other treatment modalities	
**Quality of life**	**Real-world cost-effectiveness**
No significant differences after adding TTFields to the standard protocol	Above the willingness-to-pay threshold

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
