# Peer review of "Tumor-Treating Fields in Glioblastomas: Past, Present, and Future"

_cancers, 2022, doi:10.3390/cancers14153669_

Round 1

Reviewer 1 Report

Guo and coleagues described a detailed review of tumor treating fields (TTF) in this manuscript. The mechanism of cancer cell inhibition of TTF and the clinical results of TTF to date have been well summarized. Not only can TTF increase the current standard treatment response rate of glioblastoma, but it also suggests the possibility of combination treatment along with other chemotherapy, target therapy, and immunotherapy in the future. Considering the importance of the subject covered by this manuscript, I believe that the current form can be published in Cancers.

Author Response

Thank you very much for your careful review regarding our review article “Tumor Treating Fields in Glioblastomas: Past, Present, and Future” (cancers-1810291). 

Reviewer 2 Report

The manuscript by Xiaopeng Guo and collaborators describes in detail everything that has been published and validated for the technique Tumor Treating Fields in Glioblastoma. The review is written in a clear way and objectively reports the positive and negative characteristics of this technique allowing the reader to fully explore the results obtained so far. Given that glioblastoma is one of the most aggressive tumors displaying high chemoresistance, the opportunity to apply a multimodality therapy represents an efficacious strategy.

Thus, the review is very interesting and the topic is innovative and current.

Prior publication, authors should include data from a very recent publication focused on the molecular alterations induced by this therapy (Pandey M, Xiu J, Mittal S, Zeng J, Saul M, Kesari S, Azadi A, Newton H, Deniz K, Ladner K, Sumrall A, Korn WM, Lou E. Molecular alterations associated with improved outcome in patients with glioblastoma treated with Tumor-Treating Fields. Neurooncol Adv. 2022;4(1):vdac096. doi: 10.1093/noajnl/vdac096.).

-Moreover, to make the reading even simpler, a table or figure in which the positive and negative characteristics of this technique in glioblastoma are schematized, would be useful.

-Finally, Figure 2 does not facilitate understanding because the colors used and the size of the writings are very little visible. Authors should definitely improve this by changing the colors so that the contrast makes the information easier to understand.

Author Response

Response to Reviewer 2 Comments

Point 1: The manuscript by Xiaopeng Guo and collaborators describes in detail everything that has been published and validated for the technique Tumor Treating Fields in Glioblastoma. The review is written in a clear way and objectively reports the positive and negative characteristics of this technique allowing the reader to fully explore the results obtained so far. Given that glioblastoma is one of the most aggressive tumors displaying high chemoresistance, the opportunity to apply a multimodality therapy represents an efficacious strategy. Thus, the review is very interesting and the topic is innovative and current.

Response 1: Thank you very much for your careful review regarding our article “Tumor Treating Fields in Glioblastomas: Past, Present, and Future” (cancers-1810291). In addition, our team thoroughly discussed your suggestions and comments listed below and made the recommended changes. Please see our responses below.

Point 2: Prior publication, authors should include data from a very recent publication focused on the molecular alterations induced by this therapy (Pandey M, Xiu J, Mittal S, Zeng J, Saul M, Kesari S, Azadi A, Newton H, Deniz K, Ladner K, Sumrall A, Korn WM, Lou E. Molecular alterations associated with improved outcome in patients with glioblastoma treated with Tumor-Treating Fields. Neurooncol Adv. 2022;4(1):vdac096. doi: 10.1093/noajnl/vdac096.).

Response 2: Thank you for the valuable suggestion. This is a very up-to-date publication that was released several days ago. We read this article with interest and added the reference in the manuscript. A sentence regarding this article was added under the section of “3.6 Identification of Distinct Response to TTFields Treatment” in the manuscript (lines 507-509).

Point 3: To make the reading even simpler, a table or figure in which the positive and negative characteristics of this technique in glioblastoma are schematized, would be useful.

Response 3: Thank you for the suggestion. We added a table (Table 3) to better demonstrate the positive and negative characteristics of TTFields in treating glioblastomas (line 647).

Point 4: Finally, Figure 2 does not facilitate understanding because the colors used and the size of the writings are very little visible. Authors should definitely improve this by changing the colors so that the contrast makes the information easier to understand.

Response 4: Thank you for the valuable comment. We changed the colors of Figure 2. Information conveyed in this figure is easier to read using the updated version.